# Vascular Smooth Muscle Cell Neutral Sphingomyelinase 2 in the Release of Exosomes and Vascular Calcification

**DOI:** 10.3390/ijms23169178

**Published:** 2022-08-16

**Authors:** Angelina Pavlic, Nasim Bahram Sangani, Johanna Kerins, Gerry Nicolaes, Leon Schurgers, Chris Reutelingsperger

**Affiliations:** 1Department of Biochemistry, Cardiovascular Research Institute Maastricht (CARIM), Maastricht University, 6229 ER Maastricht, The Netherlands; 2University College Maastricht, Maastricht University, 6229 ER Maastricht, The Netherlands

**Keywords:** vascular calcification, vascular smooth muscle cells, exosomes, neutral sphingomyelinase 2

## Abstract

Vascular calcification (VC) is the pathological precipitation of calcium salts in the walls of blood vessels. It is a risk factor for cardiovascular events and their associated mortality. VC can be observed in a variety of cardiovascular diseases and is most prominent in diseases that are associated with dysregulated mineral homeostasis such as in chronic kidney disease. Local factors and mechanisms underlying VC are still incompletely understood, but it is appreciated that VC is a multifactorial process in which vascular smooth muscle cells (VSMCs) play an important role. VSMCs participate in VC by releasing extracellular vesicles (EVs), the extent, composition, and propensity to calcify of which depend on VSMC phenotype and microenvironment. Currently, no targeted therapy is available to treat VC. In-depth knowledge of molecular players of EV release and the understanding of their mechanisms constitute a vital foundation for the design of pharmacological treatments to combat VC effectively. This review highlights our current knowledge of VSMCs in VC and focuses on the biogenesis of exosomes and the role of the neutral Sphingomyelinase 2 (nSMase2).

## 1. Introduction

Vascular calcification (VC) is the mineralization of vascular tissue and is considered a pathological process with high prevalence in the aging population of industrialised countries. VC is a risk factor and predictor of cardiovascular morbidity and mortality [1]. VC can develop in almost all arterial tissues and can occur in both the intimal and medial layer, also referred to as calcific atherosclerosis and Mönckeberg’s sclerosis, respectively [2]. Calcification can occur also in aortic valves [3] and small blood vessels of adipose tissue and skin (calciphylaxis) [4]. To date no pharmacological therapeutics targeting VC are available.

VC was thought to be a passive process [5] but is currently described as an active process controlled in part by vascular smooth muscle cells (VSMCs). The molecular mechanisms through which VSMCs regulate intimal and medial calcification are still not fully understood. It is clear, however, that these are closely linked to VSMC phenotypic switching in response to local cues. VSMCs can transdifferentiate across a spectrum of phenotypes from calcification-suppressing cells (contractile phenotype) into calcifying cells (osteo/chondrogenic phenotypes) [6,7]. VSMC apoptosis [8] and senescence [9] have also been demonstrated to contribute to VC.

Recent studies have highlighted the prominent role of VSMC-derived extracellular vesicles (EVs) in the regulation of VC [10,11]. Such EVs can either promote or inhibit VC dependent on the phenotype of the EV-producing cell [10]. Interestingly, EVs derived from other vascular cells such as endothelial cells can contribute to VC [12,13]. Also, platelet EVs induce VSMCs to switch towards a pro-inflammatory phenotype [14] and are thought to augment downstream processes such as VC [11]. These studies point towards a central role for EVs in VC.

EVs can be produced by different biogenetic pathways that determine their size and composition and, hence, their bioactivity. The enzyme neutral sphingomyelinase 2 (nSMase2, also known as sphingomyelin phosphodiesterase 3, SMPD3), which generates ceramide by cleaving its substrate sphingomyelin [15], has been revealed as an important enzyme in the formation of EVs and the sorting of cargo into EVs. Pharmacological inhibition of nSMase2 reduces the secretion of EVs by VSMCs and VSMC-driven calcification in vitro [16], and diminishes atherogenesis in a mouse model of atherosclerosis in vivo [17].

This review describes our current understanding of the role of VSMCs in VC and focuses on EV-biogenesis and nSMase2.

## 2. Vascular Smooth Muscle Cells in Vascular Calcification

The healthy arterial wall is composed of several cell types including VSMCs which are predominantly present in the tunica media of the vessel wall. VSMCs are highly specialized cells that maintain vascular structure and regulate vascular tone and blood pressure. VSMCs have prominent roles also in vascular pathologies such as atherosclerosis [18] and aneurysm formation [19]. They participate in early- and late-stage atherosclerosis and display great plasticity in phenotype during atherogenesis. VSMCs can adopt a wide range of phenotypes including contractile, synthetic, macrophage-like, adipocyte-like, osteogenic and stem cell-like phenotypes. These phenotypes can be distinguished by their protein-expression profiles and their abilities to contract, synthesise extracellular matrix proteins, migrate and proliferate. Literature about VSMC phenotypic switching in atherosclerosis has been excellently reviewed recently [18,20].

VSMCs have been assigned the protagonist role in our current models of the VC process. In physiology, VSMCs are predominantly in the contractile phenotype and suppress the precipitation of calcium-phosphate crystals in an environment that is supersaturated with calcium ions (Ca^2+^) and inorganic phosphate-ions (PO_4_^3−^, P_i_). On the other hand, it has been demonstrated that osteochondrogenic VSMCs, a phenotype frequently observed in vascular pathology, actively promote extracellular matrix calcification [6]. The difference in calcification-modulation between the contractile and osteochondrogenic phenotype is largely explained by differences in the expression of inhibitors and activators of calcification. Contractile VSMCs synthesise Matrix Gla Protein (MGP) which is a strong inhibitor of precipitation of calcium salts in the vascular wall and which can reverse VC [21,22]. Knocking out the *MGP* gene results in massive calcification of the aorta in vivo [21,23]. MGP is a vitamin K-dependent protein that needs to undergo post-translational gamma-carboxylation of four glutamate residues in order to express its full anti-calcification activity [24]. Interestingly, dietary intake of vitamin K lowers the levels of circulating dephosphorylated and uncarboxylated MGP, an inactive isoform of MGP which is positively correlated with the severity of VC [25] and mortality [26]. The mechanisms of action of MGP are still not fully understood but likely include inhibition of calcium-crystal growth by shielding properties and maintaining VSMCs in the contractile phenotype through blocking bone morphogenetic protein 2 (BMP-2) functions [21]. Interestingly, the in vitro phenotypic switch of contractile VSMCs by elevated Ca^2+^ is accompanied by a transient upregulation of MGP followed by a loss of MGP expression [27]. This was associated with osteogenic transdifferentiation of the VSMCs. Another potent inhibitor of calcification is extracellular pyrophosphate (P_2_O_7_^4−^, PP_i_) which is produced by ectonucleotide pyrophosphatase/phosphodiesterase-1 (eNPP1) [28]. PP_i_ can be degraded into pro-calcifying P_i_ by tissue-nonspecific alkaline phosphatase (TNAP) [29]. Hence, eNPP1 and TNAP activity regulate a balance that determines the growth of calcium crystals. This type of regulation occurs in mineralizing bone and is driven mainly by osteoblasts [30]. A similar regulation of VC by P_i_-stimulated VSMCs in aorta explants has been proposed [31]. Similar to calcifying osteoblasts that produce mineralizing matrix vesicles, pro-calcifying VSMCs generate EVs that form nucleation sites for calcification [32,33]. Pro-calcifying EVs can be released by apoptosis (apoptotic bodies, [8]), budding of vesicles from the plasma membrane (matrix-like vesicles, [34]) and fusion of multivesicular bodies (MVBs) with the plasma membrane giving release of exosomes [35]. EVs derived from VSMCs can contain Ca^2+^ and P_i_ [36], and proteins and lipids that either inhibit (MGP, Fetuin-A, prothrombin) [27,37] or stimulate calcification (annexins A1, A2 and A6, phosphatidylserine (PS), TNAP, glucose-regulated protein 78 (GRP78)) [27,38,39,40,41]. The balance between inhibitors and stimulators is believed to determine whether EVs promote calcification. Recently it was proposed that contractile VSMCs, which secrete low amounts of exosomes, respond to injury by switching to a proliferative phenotype that secretes enhanced amounts of reparative non-calcifying exosomes. Prolonged exposure to an inflammatory environment and high levels of Ca^2+^ and P_i_ push proliferative VSMCs further towards a calcifying phenotype secreting high amounts of calcifying exosomes [16]. There is, hence, a link between VSMC phenotype and the amount and composition of secreted exosomes. Recent research has focused on the biogenetic pathways of exosomes and has unveiled parts of the molecular machinery that sort cargo for loading into exosomes. Knowledge of these machineries and understanding how they operate may offer possibilities to design targeted therapies to intervene with VC. The next section reviews the literature on EVs with emphasis on exosomes.

## 3. Extracellular Vesicles: Nomenclature, Structure and Biogenesis

EVs are cell-derived particles that are encapsulated by a phospholipid bilayer. Initially, EVs were regarded as a waste-disposal system of the cell to discard superfluous and noxious material. This view has changed dramatically over the past decade. Currently, EVs are seen as important structures, which are generated by well-orchestrated processes and which serve relevant functions such as intercellular communication in physiology and pathology as for example in the developing brain [42,43,44].

Unfortunately, literature still encompasses high inconsistency regarding the nomenclature used to describe EVs. For example, EV is used to indicate exosomes and vice versa and the usage of the terms is based on the authors’ preference [44]. In order to provide guidance, the International Society for Extracellular Vesicles (ISEV) proposed a consensus nomenclature in which EV is used as a generic term to describe all lipid bilayer encapsulated particles released from cells and unable to replicate due to the lack of a functional nucleus [45]. EV subtypes are distinguished on the basis of biogenesis, size, composition and mechanism of release [46]. Exosomes are the smallest vesicles with a size from 30 nm to 100 nm in diameter. Microvesicles range in diameter from 100 nm to 1000 nm and apoptotic bodies have diameters larger than 500 nm. EV subclasses can be isolated by methods that separate on size, density, and antigen expression [46]. Overlapping characteristics between the different subtypes have hampered the assignment of unique parameters to either subtype and, consequently, an exact description of the EV subtype and its biogenetic pathway, features and bioactivity are not possible. Hence, one has to realise that published research on EVs has been performed with mixtures of EVs that are enriched for a specific EV subclass. Apoptotic bodies have a broad size distribution and result from a process that orchestrates the demise of the cell and that produces lipid membrane delimited cellular fragments containing a broad variety of cellular components including organelles, proteins, DNA and RNA [47,48]. Apoptotic bodies are removed from the tissue by phagocytosis [49]. Unphagocytosed apoptotic bodies of VSMCs have been shown to stimulate calcification [8]. The second subclass of EVs, the microvesicles, are generated from plasma membrane segments by outward budding and regulated “pinching off” of a vesicular membrane structure, a process indicated with the term scission. Microvesicle formation requires cytoskeletal actin and microtubules, molecular motors such as dynein, kinesis and myosin, soluble N-ethylmaleimide sensitive factor (NSF) attachment protein receptors (SNAREs), Ras-associated binding proteins (Rab) guanosine triphosphatases (GTPases) and tethering factors [50,51]. Microvesicles produced under pathological conditions such as atherosclerosis can enhance calcification [34]. The third subclass of EVs, the exosomes, are the smallest EVs and they are formed by the complex multistep endocytic membrane transport pathway. This pathway starts with the formation of early endosomes by invagination and inward budding and scission of the plasma membrane. Early endosomes evolve into late endosomes and multivesicular bodies (MVBs) carrying small lipid bilayer encapsulated vesicular structures that are termed intraluminal vesicles (ILVs) [46]. Endosome maturation is mainly mediated by the small GTPases Rab5 and Rab7 in a process called Rab conversion. Rab5, the marker of early endosomes, is replaced by Rab7, the marker of late endosomes [52,53]. Once formed, MVBs can either fuse with lysosomes for lysosomal degradation or with the plasma membrane for secretion of the ILVs as exosomes [54]. Recently it was demonstrated that nSMase2-activity increases exosome secretion by inhibiting disrupting Vacuolar-type ATPase (V-ATPase) assembly on MVB and inhibiting, consequently, acidification of MVBs and MVB sorting towards lysosomes [55]. Once rescued from lysosomal degradation by nSMase2-activity, MVBs can dock to the plasma membrane to release their exosomes. Rab27a and Rab27b have been shown to play a key role in exosome secretion through targeting MVBs to the cell periphery and their docking at the plasma membrane [56].

The cargo that exosomes carry does not arise from a stochastic process but is determined by sorting mechanisms closely linked to the ILV biogenetic pathways. ILVs are generated by inward budding and scission of the limiting membranes of late endosomes through two distinct pathways: (i) the well-studied endosomal sorting complexes required for transport (ESCRT) dependent pathway and (ii) the ESCRT independent pathway. The ESCRT dependent pathway is driven by a protein machinery consisting of four complexes (ESCRT-0,-I,-II,-III) which are sequentially recruited to the limiting membrane of the late endosome and which cause sorting of ubiquitinated proteins, inward budding and scission [54,57,58]. Depletion of key ESCRT subunits could not fully inhibit MVB formation and exosome secretion demonstrating existence of ESCRT-independent pathways [59]. The ESCRT-independent pathway is driven by ceramide, which is generated in the limiting membrane by the action of the enzyme nSMase2 and which promotes inward budding [60] (see also next section). The ESCRT-independent pathway selects cargo by the sorting properties of ALG-2-interacting protein X (ALIX) [61] and the microtubule-associated protein 1A/1B-light chain 3 (LC3), a key component of the autophagy machinery [62]. Interestingly, VSMCs submitted to calcifying conditions switch phenotype and upregulate the release of calcifying exosomes which depends on the activity of nSMase2 [16,37]. Proteomic analyses of these exosomes reveal a composition that contains both pro- and anti-calcifying proteins, the balance of which determines their propensity to induce and propagate calcification. Kapustin et al. demonstrated that part of the cargo was derived from endocytosis of extracellular components [16,37] (Figure 1). Their experiments also strongly suggest that switching of contractile VSMCs towards the pro-calcifying phenotype is accompanied by upregulation of nSMase2-dependent secretion of pro-calcifying exosomes.

## 4. Neutral Sphingomyelinase 2: Structure and Function in Exosome Release

Sphingomyelinases (SMases) are intracellular enzymes that catalyse the formation of ceramide by hydrolysis of sphingomyelin. SMases can be classified into acid, neutral and alkaline SMases on the basis of their respective optimal pH for the expression of enzymatic activity. Mammalian neutral SMases (nSMases) can be categorised into 4 types: nSMase1, nSMase2, nSmase3 and mitochondria-associated nSMase (MA-nSMase) [63]. The different nSMases are believed to function in different cellular compartments and to support different cellular functions [64,65,66]. For example, nSMase1 is associated with endoplasmic reticulum (ER) and nucleus [67] and believed to be important for ceramide production during ER-stress [63], nSMase2 is predominantly localised to the plasma membrane, and to the membranes of the Golgi and the endosomal recycling compartments [68,69] and crucial for the production of ceramide in support of ILV-formation of MVBs [60]. MA-nSMase is detected at mitochondria-associated membranes [70] and believed to participate in apoptotic pathways [63,71]. The catalytic activity of nSMases is enhanced by divalent cations such as magnesium ions (Mg^2+^). The negatively charged aminophospholipid PS activates all nSMases but has no effect on nSMase1 activity [72]. Except for a set of conserved residues of the catalytic site, suggesting that nSMases catalyse hydrolysis of sphingomyelin by a common mechanism, the 4 types of nSMases share little structural homology [63].

Human nSMase2 is encoded by the *SMPD3* gene and has a single polypeptide chain of 655 amino acids which is organised in functionally distinct domains. The polypeptide chain has an N-terminal region (residues 1–84) containing two hydrophobic segments, a collagenous domain (residues 119–340) and a C-terminal catalytic domain (residues 341–655) [69]. Tagging experiments indicated that the two hydrophobic segments are inserted into the plasma membrane without spanning the entire membrane [73] (Figure 2).

The N-terminal region harbouring the two hydrophobic segments is necessary for PS binding [75]. The N-terminus, the catalytic domain and the C-terminus are located at the cytosolic side of the plasma membrane. This topology poses an interesting problem in the light of our understanding that its substrate sphingomyelin is preferentially located in the outer plasma membrane leaflet. During apoptosis lipid scrambling moves sphingomyelin from the outer to the inner leaflet where it is cleaved by SMases [76]. Whether a similar mechanism operates during nonapoptotic ceramide production by nSMase2 remains unknown to date. Recently the crystal structure of the catalytic domain of human nSMase2 was elucidated at 1.85-Å resolution revealing that the region connecting the catalytic domain with the N-terminal domain contains a binding site for the positive allosteric effector PS [74,77]. The widely used non-competitive inhibitor GW4689 of nSMase2 exerts its inhibitory activity through competing with PS for binding to the allosteric site of nSMase2 [74,78].

Ceramide generated by nSMases can activate various intracellular signalling pathways including the apoptotic cascade [79,80]. For example, apolipoprotein C-I (ApoC-I) activates nSMases of human aortic VSMCs resulting in increased production of ceramide, which can mediate cytochrome C-release, procaspase 3 activation and subsequently apoptosis [81]. A causal connection between nSMase-produced ceramide, apoptosis and calcification was shown with OxLDL stimulated human femoral artery VSMCs [82]. In addition, ceramide produced by nSMases can activate a nonapoptotic pathway towards calcification. It causes clustering of cholesterol-rich domains [83] and bending of the phospholipid bilayer with a negative curvature as a consequence of its cone-shaped structure [84]. Trajkovic et al. were the first to demonstrate that this property of ceramide is involved in biogenesis of MVBs by mediating ILV formation in a manner that does not depend on the ESCRT-machinery [60]. The properties of ceramide to coalesce cholesterol-rich microdomains and to bend the phospholipid membrane are believed to drive ILV formation. Using the nSMase inhibitors GW4689 [78] and spiroepoxide [85], and RNA interference the authors were able to show that the ceramide necessary for ILV formation and exosome secretion was produced by nSMase2. This landmark study on ceramide’s role in exosome secretion triggered numerous investigations into the role of sphingolipids and sphingomyelinases in EV biogenesis (for recent review see [86]. Many studies utilised GW4869 and spiroepoxide to show a role for nSMase2. The selectivity and efficacy of these inhibitors, however, have never been demonstrated unambiguously. It has been reported that GW4869 had no effect on the secretion of exosomes by several cancer cell lines whereas nSMase2 knockout had a profound inhibitory effect on these cells [87]. Hence, inhibitors cannot be used alone to prove a role for nSMase2. Methods should also be included that inactivate the *SMPD3* gene (CRISPR/Cas9) or interfere with nSMase2 mRNA translation (shRNA, siRNA). Table 1 lists a series of published studies that utilised these methods to demonstrate the involvement of nSMase2 in the secretion of exosomes by cultured cells.

Genetic deficiencies of nSMase2 by full knockout or local knockdown strategies underscored the relevance of nSMase2 to exosome secretion in vivo. Dinkins et al. crossed the 5XFAD mouse, which is a model of Alzheimer’s Disease, with the m-smpd3*^fro/fro^* mouse, which is deficient in functional nSMase2 [98]. The *fro*;5XFAD mice produced significantly less exosomes in the brain as compared to the 5XFAD mice [95]. Dickens and co-workers showed that intrastriatal injection of IL-1β in C57BL/6J mice caused neutrophil influx which was absent in m-smpd3*^fro/fro^* mice [99]. The authors did not measure the effect of IL-1β on exosome secretion in situ but were able to demonstrate convincingly that the neutrophil influx was invoked by an nSMase2-dependent secretion of exosomes by astrocytes. Lecuyer et al. performed intracortical injections of CRISPR-Cas9 constructs designed to inactivate the *SMPD3* gene in the mouse brain. Inactivation of the *SMPD3* gene suppressed the secretion of exosomes by microglia as demonstrated in vitro after isolation of microglia from brain tissue [100]. They also studied the uptake of exosomes by isolated microglia and found that diminishing nSMase2 activity changed recognition of exosomes suggesting involvement of nSMase2 in cargo selection. A comparable conclusion was reached by Guo and coworkers who found that nSMase2 was involved in packaging of prion protein into exosomes [93]. Leidal et al. who investigated secretory autophagy in HEK293T cells, proposed an LC3-dependent EV loading and secretion (LDELS) process [62]. This pathway depends on the activity of nSMase2 as demonstrated by nSMase2 knockdown. Employing proximity-dependent biotinylation proteomics the authors revealed that LC3 recruits factor associated with neutral sphingomyelinase activation protein (FAN) to the limiting membrane where it is required for ILV-formation, likely by activating nSMase2 [15]. LC3 also captures proteins such as the RNA binding proteins scaffold-attachment factor B (SAFB) and heterogeneous nuclear ribonucleoprotein K (KHNRNPK), which are then loaded into the ILVs that get secreted as exosomes. This study is the first to provide a link between nSMase2 and a molecular machinery for cargo selection (Figure 1).

## 5. Future Perspectives

Recent studies have unveiled nSMase2 as one of the key enzymes in exosome secretion by a broad range of cell types. nSMase2-activity participates in biogenesis, cargo selection and the fate of MVBs and, consequently, in the bioactivity of exosomes. It was demonstrated that nSMase2-activity is required for the release of pro-calcifying exosomes by VSMCs. Hence, nSMase2 is a potential target for treating VC pharmacologically. Currently, only few inhibitors have been described to target nSMase2 of which PDCC seems to have the most favourable features from a pharmacological perspective [101]. PDDC, a noncompetitive inhibitor, was the result of chemical optimization of the main hit from a human nSMase2 high throughput screen. Whether PDDC has satisfactory selectivity remains to be determined. The availability of the crystal structure of the catalytic domain of human nSMase2 has opened possibilities to discover superior competitive inhibitors by virtual screening strategies. Inhibitors targeting nSMase2 will face great challenges on their route towards their application in the pharmacological treatment of patients with VC because of the anticipated ubiquity of the nSMase2-exosome axis in the patient. The anticipated major challenge will be the effects of inhibitors on the brain and the bone since nSMase2 plays prominent roles both in brain and bone physiology [102,103]. Targeted delivery of inhibitors to vascular tissue would be a strategy to reduce potential adverse effects of nSMase2-inhibitors on brain and bone. Such strategy is considered feasible since targeting inflamed blood vessel walls with nanostructures containing pharmacons has demonstrated efficacy in treating atherosclerotic lesions of the aorta in mouse models [104,105].

## Figures and Tables

**Figure 1 ijms-23-09178-f001:**
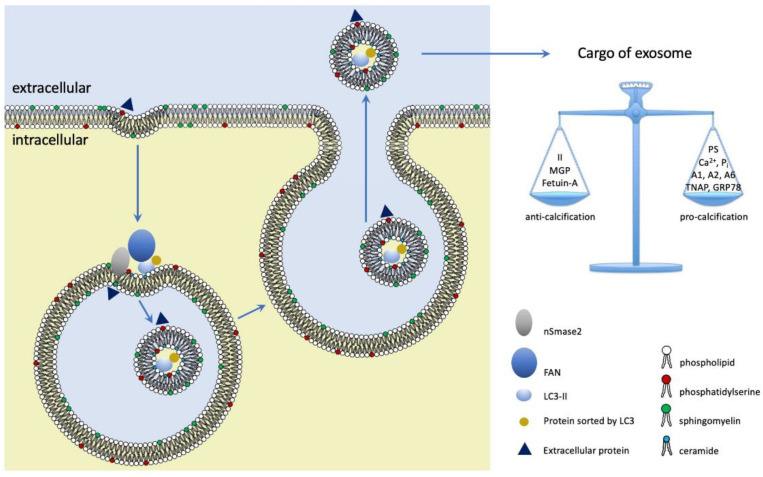
The nSMase2-dependent pathway of ILV-formation in MVBs. nSMase2 produces ceramide which bends the membrane inward by its cone-shaped structure. nSMase2 is likely activated by FAN, which is recruited to the limiting membrane by LC3, which also selects cargo for the ILVs. VSMCs can secrete exosomes that either stimulate or inhibit VC depending on the balance of their cargo. Cargo is composed of endocytosis of extracellular compounds and sorting of intracellular compounds during ILV formation. Whether LC3 is involved in sorting anti- and pro-calcifying compounds is not known to date. ILV: intraluminal vesicle. MVB: multivesicular body. II: prothrombin. PS: phosphatidylserine. A1, A2, A6: annexins A1, A2 and A6. TNAP: tissue non-specific alkaline phosphatase. GRP78, glucose-regulated protein 78,000.

**Figure 2 ijms-23-09178-f002:**
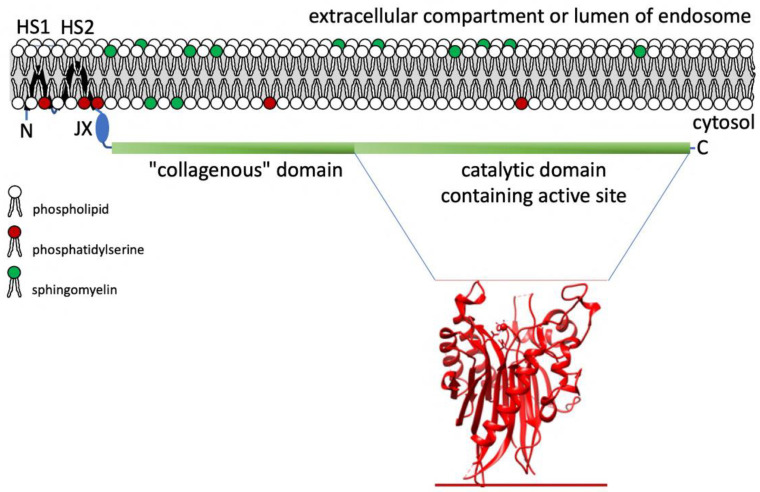
Membrane topology of nSMase2 and tertiary structure of the C-terminal part of the catalytic domain. The domain organisation and topology of the domains are derived from [73]. The tertiary structure is based on coordinates taken from [74]. N: N-terminus. HS1 and HS2: Hydrophobic segments 1 and 2. JX: domain juxtapositioned to the domain with the hydrophobic segments and critical for allosteric activation. C: C-terminus.

**Table 1 ijms-23-09178-t001:** Studies that examined the role of nSMase2 in exosome composition and release by cells in culture. TEM: Transmission Electron Microscopy, WB: Western Blotting, NTA: Nanoparticle Tracking Analysis, siRNA: small interfering RNA, shRNA: short hairpin RNA, miR: micro RNA.

Cell Type	Exosome Verification Method	Method to Demonstrate Role of nSMase2	Examined Cargo of the Exosomes	Reference
Oli-neu	TEM	GW4869, spiroepoxide, siRNA	ProteoLipid Protein	[60]
HEK293	WB (CD63)	GW4869, siRNA, overexpression	miR-16, miR-146a	[88]
Neuro2A	TEM, WB (Alix, Tsg101)	GW4869, siRNA	pro-Aβ fibrillogenesis activity	[89]
THP-1	TEM, WB (CD63)	GW4869, spiroepoxide, shRNA	anti-viral activity	[90]
MDA-MB-231	SEM, WB (CD63)	GW4869, overexpression	miR-106	[91]
Primary human VSMCs	NTA, WB (CD9, CD63)	GW4869, spiroepoxide, siRNA	pro-calcifying activity	[16]
Primary murine microglia	IEM (Tsg101)	GW4869, siRNA	Tau46	[92]
GT1-7	TEM, WB (Tsg101, Flotillin-1)	GW4869, RNAi	Prion protein	[93]
Primary human cardiosphere-derived cells	TEM, NTA, WB (CD63, HSP70)	siRNA	pro-angiogenic and pro-survival activity	[94]
Primary mouse astrocytes	TNA, WB (Alix, Tsg101)	m-nSMase2*^fro/fro^*	Aβ oligomers	[95]
SKBR3	TEM, NTA, WB (Alix, Tsg101, CD81)	GW4869, siRNA	Hsc70	[96]
PC3	TEM, NTA, WB (CD63)	CRISPR/Cas9	PD-L1	[87]
TIG-3	TEM, NTA	siRNA, overexpression	none studied	[97]
HEK293T	TEM, WB (Alix, Tsg101, CD9)	GW4869, shRNA	LC3-II, SAFP, HNRNPK	[62]
Hela	NTA, WB (Alix, CD63, CD81, syntenin)	GW4869, siRNA	V-ATPase transmembrane subunit	[55]

## Data Availability

Not applicable.

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
