# Peer review of "Vascular Smooth Muscle Cell Neutral Sphingomyelinase 2 in the Release of Exosomes and Vascular Calcification"

_ijms, 2022, doi:10.3390/ijms23169178_

Round 1
Reviewer 1 Report
The authors have provided a clearly written and organized review on the role of nSM2 in vascular calcification. There are a few minor suggestions for edits that would form a more complete review of the subject.
1) The product of nSM2, ceramide, has effects independent of its role in EV formation. Some specific details describing ceramide effects that may directly or indirectly influence pathways that regulate vascular calcification would deepen the readers understanding of nSM2 in this area of vascular pathophysiology.
2) EVs derived from cell types other than VSMCs have been reported (i.e. endothelial cells). Consideration of these findings would enhance the readers knowledge in this field.
Author Response
We thank the reviewer for his/her constructive and valuable suggestions. We have adapted the text accordingly.
1) The product of nSM2, ceramide, has effects independent of its role in EV formation. Some specific details describing ceramide effects that may directly or indirectly influence pathways that regulate vascular calcification would deepen the readers understanding of nSM2 in this area of vascular pathophysiology.
Lines 240 - 247
“Ceramide generated by nSMases can activate various intracellular signaling pathways including the apoptotic cascade (Nikolova-Karakashian and Rozenova 2010; Stith et al. 2019). For example, Apolipoprotein C-I (ApoC-I) activates nSMases of human aortic VSMCs resulting in increased production of ceramide, which can mediate cytochrome C-release, procaspase 3 activation and subsequently apoptosis (Kolmakova et al. 2004). A causal connection between nSMase-produced ceramide, apoptosis and calcification was shown with OxLDL stimulated human femoral artery VSMCs (Liao et al. 2013). In addition, ceramide produced by nSMases can activate a nonapoptotic pathway towards calcification.”
2) EVs derived from cell types other than VSMCs have been reported (i.e. endothelial cells). Consideration of these findings would enhance the readers knowledge in this field.
Lines 49 - 53
“Interestingly, EVs derived from other vascular cells such as endothelial cells can contribute to VC (Mas-Bargues et al. 2022; Lin et al. 2022). Also, platelet EVs induce VSMCs to switch towards a pro-inflammatory phenotype (Vajen et al. 2017) and are thought to augment downstream processes such as VC (Schurgers et al. 2018). These studies point towards a central role for EVs in VC.”
Reviewer 2 Report
With the knowledge that vascular calcification is an important risk factor for cardiovascular events and mortality, this paper reviews the roles of vascular smooth muscle cells and neutral sphingomyelinase 2 in vascular calcification. As this key enzyme is needed for the release of pro-calcifying exosomes by vascular smooth muscle cells, its inhibition could be a potential target for the pharmacological treatment of vascular calcification.
The topic is important and the manuscript provides a comprehensive analysis of the subject. I would recommend this manuscript for publication after the following suggestions have been attended to:
All acronyms should be defined when first used
Line 36: sentence starting with “Calcification...” needs citation
Lines 79-87: I suggest adding these important ideas:
- MGP is a strong inhibitor and the only factor that can reverse VC (doi: 10.3389/fmed.2020.00154).
- Circulating plasma dephosphorylated-uncarboxylated MGP (dp-ucMGP), associated with VC severity, is lowered by vitamin K intake (doi: 10.3390/antiox10040566).
- Higher dp-ucMGP is likely causally linked with total and cardiovascular mortality (doi: 10.1161/HYPERTENSIONAHA.114.04494).
Line 182: “Proteomic analysis of these exosomes reveals” or “Proteomic analyses of these exosomes reveal”
Lines 191-207: I suggest citing DOI: 10.1042/BJ20101752 (“only overexpression of nSMase2, but not nSMase1 or nSMase3, had significant effects on cellular sphingolipid levels, increasing ceramide and decreasing sphingomyelin”) and DOI: 10.1016/j.redox.2014.07.006
Line 243: please revise the sentence starting with “Therefore...”
Figure 2 is not mentioned in the text
Author Response
We thank the reviewer for his/her constructive and valuable comments and suggestions.
All acronyms should be defined when first used
Done
Line 36: sentence starting with “Calcification...” needs citation
Lines 36 - 37
“Calcification can occur also in aortic valves (Lindman et al. 2021) and small blood vessels of adipose tissue and skin (calciphylaxis) (Brandenburg et al. 2012).”
Lines 79-87: I suggest adding these important ideas:
- MGP is a strong inhibitor and the only factor that can reverse VC (doi: 10.3389/fmed.2020.00154).
- Circulating plasma dephosphorylated-uncarboxylated MGP (dp-ucMGP), associated with VC severity, is lowered by vitamin K intake (doi: 10.3390/antiox10040566).
- Higher dp-ucMGP is likely causally linked with total and cardiovascular mortality (doi: 10.1161/HYPERTENSIONAHA.114.04494).
Lines 86 - 93
“Contractile VSMCs synthesize Matrix Gla Protein (MGP) which is a strong inhibitor of precipitation of calcium salts in the vascular wall and which can reverse VC (Schurgers et al. 2013; Roumeliotis et al. 2020). Knocking out the MGP gene results in massive calcification of the aorta in vivo [16,17]. MGP is a vitamin K-dependent protein that needs to undergo post-translational gamma-carboxylation of four glutamate residues in order to express its full anti-calcification activity [18]. Interestingly, dietary intake of vitamin K lowers the levels of circulating dephosphorylated and uncarboxylated MGP, an inactive isoform of MGP which is positively correlated with severity of VC (Popa et al. 2021) and mortality (Liu et al. 2015).”
Line 182: “Proteomic analysis of these exosomes reveals” or “Proteomic analyses of these exosomes reveal”
Lines 194 - 195
“Proteomic analyses of these exosomes reveal”
Lines 191-207: I suggest citing DOI: 10.1042/BJ20101752 (“only overexpression of nSMase2, but not nSMase1 or nSMase3, had significant effects on cellular sphingolipid levels, increasing ceramide and decreasing sphingomyelin”) and DOI: 10.1016/j.redox.2014.07.006
Lines 207 - 209
“The different nSMases are believed to function in different cellular compartments and to support different cellular functions (Xiang et al. 2021; Clarke et al. 2011; Moylan et al. 2014).“
Line 243: please revise the sentence starting with “Therefore...”
Lines 261 - 264
“Hence, inhibitors cannot be used alone to prove a role for nSMase2. Methods should also be included that inactivate the SMPD3 gene (CRISPR/Cas9) or interfere with nSMase2 mRNA translation (shRNA, siRNA).”
Figure 2 is not mentioned in the text
Lines 225 - 226
“Tagging experiments indicated that the two hydrophobic segments are inserted into the plasma membrane without spanning the entire membrane [64] (Figure 2).”